# A novel generative framework for designing pathogen-targeted antimicrobial peptides with programmable physicochemical properties

**Weizhong Zhao**[1,2,3]*, **Kaijieyi Hou**[1,2,3], **Chang Tang**[1,2,3], **Yiting Shen**[4], **Jinlin Liu**[5], **Xiaohua Hu**[6]*

**1** Hubei Provincial Key Laboratory of Artificial Intelligence and Smart Learning, Central China Normal University, Wuhan, Hubei, China, **2** School of Computer, Central China Normal University, Wuhan, Hubei, China, **3** National Language Resources Monitoring and Research Center for Network Media, Central China Normal University, Wuhan, Hubei, China, **4** Detroit Green Technology Institute, Hubei University of Technology, Wuhan, Hubei, China, **5** School of Life Sciences, Central China Normal University, Wuhan, Hubei, China, **6** College of Computing and Informatics, Drexel University, Philadelphia, Pennsylvania, United States of America

* wzzhao@ccnu.edu.cn (WZ); xh29@drexel.edu (XH)

## Abstract

Antimicrobial peptides (AMPs) are crucial in addressing the global crisis of bacterial resistance. However, there are still significant limitations in existing methods on de novo AMPs design, especially in designing AMPs with desirable physicochemical properties for specific bacterial pathogens. In this study, we propose a novel generative framework for designing pathogen-targeted antimicrobial peptides with programmable physicochemical properties. More specifically, a conditional Variational Autoencoder is first pretrained for generating AMPs with editable physicochemical properties. We then develop a conditional diffusion model to learn hidden representations of AMPs for targeting pathogens of interest, and construct corresponding MIC predictors for specific bacterial strains. Through comprehensive simulation experiments, we demonstrate that the proposed framework outperforms most existing models in terms of antimicrobial efficacy against specific bacterial targets. Moreover, through systematic screening and analysis, we have identified two star AMPs for each of the two target bacterial species (i.e., *E. coli* or *S. aureus*), both of which exhibit excellent performance in antibacterial activity, hemolytic properties, toxicity profiles, etc. Overall, this study provides the key technological support for developing next-generation intelligent platforms for antimicrobial agents design.

## Author summary

In our study, we propose a novel generative framework for designing antimicrobial peptides (AMPs) with programmable physicochemical properties and targeting capabilities. AMPs are crucial tools in addressing the global bacterial resistance

**Data availability statement:** The data and source codes are available in GitHub at https://github.com/David-WZhao/AMPGen.

**Funding:** This work is supported by the National Natural Science Foundation of China (62472192 to [WZ]; 62372205 to [WZ]); the Fundamental Research Funds for the Central Universities (KJ02502022-0450 to [WZ]); the National Language Commission Key Research Project (ZDI145-56 to [WZ]); the Self-determined Research Funds of CCNU from the Colleges' Basic Research and Operation of MOE (CCNU25JC008 to [WZ]). The funders had no role in study design, data collection and analysis, decision to publish, or preparation of the manuscript.

**Competing interests:** The authors have declared that no competing interests exist.

crisis, but existing design methods still have significant limitations, especially when designing AMPs with ideal physicochemical properties for specific pathogens. We utilize a conditional Variational Autoencoder (CVAE) to generate AMPs with adjustable physicochemical characteristics and further develop a conditional diffusion model to learn the hidden representations of AMPs for precise pathogen targeting. Additionally, we construct corresponding minimum inhibitory concentration (MIC) prediction models for different bacteria to evaluate the antimicrobial efficacy of the generated AMPs. Through extensive simulation experiments, we demonstrate that this framework outperforms most existing models in terms of efficacy against specific bacterial targets. Our research provides key technological support for developing next-generation intelligent platforms for antimicrobial agent design, with significant practical applications.

## 1 Introduction

Antibiotics, which are capable of inhibiting or eliminating bacterial pathogens, have long served as the cornerstone of therapeutic strategies targeted against pathogens [1,2]. However, the development of novel antibiotics has stagnated globally, with no completely original antibiotic structural classes emerging in nearly four decades [3]. This is largely attributable to the rapid development of antimicrobial resistance by ESKAPE pathogens (i.e., *Enterococcus faecium*, *Staphylococcus aureus*, *Klebsiella pneumoniae*, *Acinetobacter baumannii*, *Pseudomonas aeruginosa*, and *Enterobacter spp.*) under in vitro conditions, which significantly compromises antibiotic efficacy [4–6]. Notably, antimicrobial peptides (AMPs) have demonstrated remarkable efficacy in overcoming such resistance mechanisms, making the discovery and development of novel AMPs an urgent priority in modern antimicrobial research [7,8].

AMPs, typically composed of 5-50 amino acids, exhibit unique structural configurations and functional versatility that enable them to disrupt bacterial membranes for effective bactericidal activity [9,10]. Moreover, this membrane-targeting mechanism demonstrates an evolutionary advantage by effectively reducing the likelihood of resistance development compared to conventional antibiotics [11,12]. Nevertheless, challenges persist in large-scale clinical implementation due to intricate synthesis protocols, prohibitive manufacturing costs, and the delicate balance required between antimicrobial potency and structural stability optimization. All the above factors result in protracted development timelines and suboptimal translational efficiency [13].

Recent years have witnessed the emergence of sophisticated computational approaches for AMPs design, spanning three primary methodologies: data mining for putative AMP sequences [11,14–16], virtual modification of peptide sequences analogous to computer directed evolution [17] and de novo generation through generative models (e.g., VAE (Variational Autoencoder)-based [18–20], Transformer-based [21], GAN (Generative Adversarial Network)-based [22,23], conditional VAE-based [24–26] and diffusion-based [27,28] models) or fine-tuned large language models [29]. While these models demonstrate remarkable efficacy in identifying broad-spectrum

AMPs via high-throughput generation and screening, the systematic evaluation reveals persistent technical barriers in engineering AMPs with prespecified functionality, particularly those requiring precise control over targeted physicochemical properties or pathogen-specific bactericidal activity.

In this paper, we effectively integrate conditional VAE (CVAE) and hidden diffusion model to develop a dual-functionality framework that can generate AMPs with expected physicochemical properties [26] (including molecular weight, isoelectric point (pI), gravy, aromaticity, instability index, disulfide bonds, molecular volume, and secondary structure fraction ($\alpha$-helix, $\beta$-sheet, and random coil)) and targeting specific bacteria (e.g., *E. coli, S. aureus*). In order to better evaluate the antibacterial effect of AMPs against target bacteria, we also develop a tool for predicting the minimum inhibitory concentration (MIC) of different bacteria. The framework implements a multi-phase training architecture (Fig 1A) through systematically engineered computational pipelines. In the initial phase, we curate a dataset of 2.3 million amino acid sequences (with less than 50 residues) from UniProt database [30] through stringent length-based filtration. These sequences undergo comprehensive physicochemical profiling via BioPython's ProteinAnalysis toolkit to establish sequence-property correlations for conditional VAE (CVAE) pre-training. Subsequently, a subset of 480,358 sequences is processed through the transformer-based encoder module of the pre-trained CVAE to generate latent representations for AMPs space, which served as the foundational dataset for hidden diffusion model pre-training. Thereafter, to enable pathogen-targeted AMPs

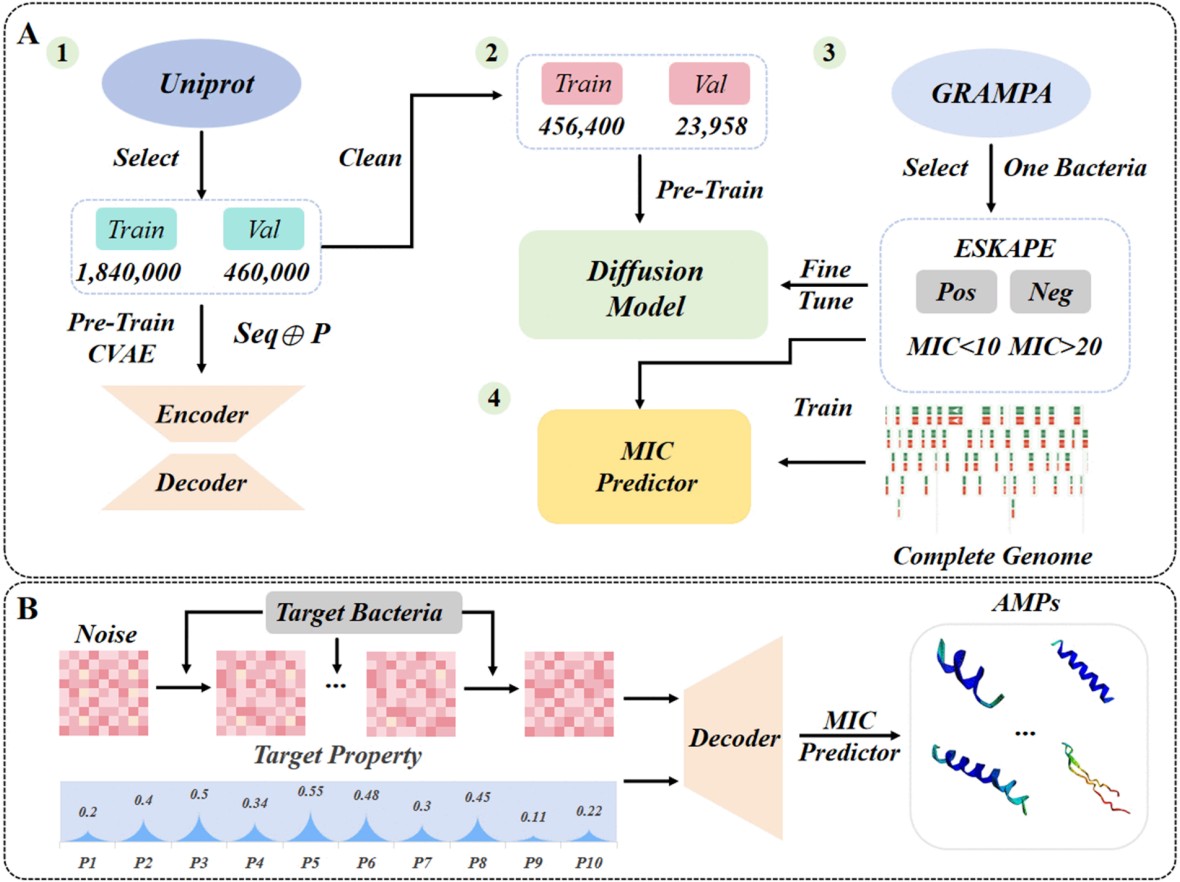

**Fig 1. Schematic representation of the overall framework.** (A) Training phase: 1. Pre-trained Conditional VAE. The symbol $\oplus$ represents the concatenation operation. 2. Pre-trained hidden diffusion model. 3. Fine-tuning the conditional diffusion model. 4. Trained MIC predictor. (B) Generating phase: AMPs generation and MIC prediction.

generation, we implement strategic model fine-tuning using curated AMPs from the Giant Repository of AMP Activities (GRAMPA) dataset [31]. These AMPs are stratified into high-efficacy and low-efficacy categories based on corresponding MIC values against specific pathogenic bacteria, enabling discriminative training of the conditional diffusion model. Finally, we develop a robust MIC prediction module through supervised learning on this efficacy-classified dataset. When generating (Fig 1B), we first sample the noise data from a standard normal distribution (i.e., Gaussian noise), process it through the conditional diffusion model, obtain the target-specific AMP's hidden representation, and then pass it through the CVAE Decoder together with the expected physicochemical properties to obtain the final AMP sequences.

To evaluate the proposed framework, we have trained generative models for targeting two representative types (i.e., *E. coli* and *S. aureus*) of ESKAPE pathogens based on our framework. After multiple experimental analysis, our model can generate AMPs with expected physicochemical properties and target specificity. Moreover, we have specifically selected two star AMPs for each of the two targeted bacterial species. These candidate peptides demonstrate substantial application potential, exhibiting not only high antibacterial efficacy but also low hemolytic activity and near-zero toxicity. Notably, our dual-functionality framework not only enables customized AMP generation but also provides a systematic platform for elucidating relationships between physicochemical properties and critical biological outcomes, particularly membrane-disruptive mechanisms and cytotoxic profiles. This study offers unprecedented opportunities for mechanistic exploration of AMP action modalities through controlled parametric analyses. Generally, the proposed framework for AMP customization and analysis provides the potential to contribute to combating the antibiotic resistance crisis globally.

The main contributions of this study are summarized as follows:

- **Dual-model collaborative generation framework.** We construct a two-stage generation framework in which a CVAE and a diffusion model work collaboratively.
- **Multi-stage large-scale pretraining and finetuning strategy for high-quality latent space construction.** We design a staged training pipeline that progressively builds a high-quality latent representation space.
- **Pathogen-specific discriminative conditional training mechanism.** AMPs are grouped into positive and negative classes based on their MIC against specific bacteria. These labels are then injected as conditional signals into the diffusion model to enable discriminative training.
- **Integrated MIC prediction module enabling a design–evaluation closed loop.** In addition to the generative framework, we train a supervised MIC predictor that estimates the MIC of candidate peptides against different bacteria, thereby quantifying their expected antibacterial potency.

## 2 Material and methods

### 2.1 Data preparation

#### 2.1.1 Pre-train data.
To enable the model to fully learn the essential semantics between sequences and physicochemical properties, we obtained 613,837 protein entries from the reviewed version of the UniProt database (in 12/17/2024 with the source file entitled "uniprotkb_reviewed_true_2024_12_17.fasta"). We used a variable-length, non-overlapping sliding window to segment each protein sequence into short fragments of 2-50 amino acids, since that lengths of most AMPs in training dataset fall in the range [2,50]. In order to achieve an approximately uniform length distribution, we set the total number of segments to 6,138,370 (ten times of protein entries from the UniProt database) and evenly distribute the target number (about 125,273) of segments for each length between 2 and 50. Specifically, during segmentation, we randomly select a length $L$ from the range [2, 50] as the window length. If the accumulated number of segments for the current length exceeds the target number, a new length will be randomly selected again. If the length of the remaining sequence is shorter than the selected $L$ when the end of the original sequence is reached, the remaining part is kept as it is. Finally, these sequences were segmented according to the AMP lengths, deriving 6,175,893 sequences. After deduplication, 5,830,424 unique sequences were obtained.

After analyzing ten physicochemical properties [26], we filtered and obtained 2,300,000 training pairs. Note that the properties used in this study are all obtained through a procedure of uniform normalization. Due to the limit of space, the advantage of pretraining on large-scale datasets is provided in S3 Appendix, and the normalized maximum and minimum values for each property are shown in S5 Appendix.

**2.1.2 Fine-tune data.** We defined the MIC thresholds based on the fact that a lower MIC indicates stronger antibacterial activity [32]. To obtain two activity classes with clear separation and comparable sample sizes for stable discriminative training, peptides with MIC <10 $\mu$M were labeled as positive (i.e., highly active) and peptides with MIC >20 $\mu$M were labeled as negative (i.e., weakly active), while peptides with intermediate MIC values were excluded [33]. Positive samples and negative samples for fine-tuning the conditional diffusion model were obtained from GRAMPA. From 8,049 entries, 4,546 unique sequences were retained. For AMPs targeting *E. coli*, the training dataset includes 2,073 positive samples and 1,969 negative samples, while for AMPs targeting *S. aureus*, the training dataset includes 1,921 positive samples and 1,717 negative samples.

**2.1.3 MIC data.** The AMP-MIC pairs for training the MIC predictor were obtained from GRAMPA as well. Genomic data for *E. coli* and *S. aureus* were obtained from NCBI [34].

## 2.2 Conditional variational AutoEncoder

**2.2.1 Model structure.** In this study, the CVAE adopts our prior Transformer-based architecture (featuring encoder-decoder layers)[26], a design extensively validated in benchmarks [25,35] (Fig 2A). A critical architectural modification involves replacing the fully connected layers previously used for latent variable generation with convolutional operations to enhance feature extraction efficacy. For training CVAE, the input properties (i.e., the expected or pre-designed physicochemical properties) are first processed via a normalization procedure. During the encoding phase, these normalized properties are jointly embedded with sequence representations through dedicated fusion layers. As for the decoding stage, the latent variables are dynamically integrated with the property information via cross-attention mechanisms. This bidirectional integration framework enables the model to establish robust correlations between sequence patterns and their physicochemical properties. A more detailed description of the model is provided in S1 Appendix.

**2.2.2 Loss function.** Beyond the standard reconstruction and KL divergence losses in the VAE framework, we introduce an property preservation loss to control the physicochemical properties of generated AMPs. To enable gradient propagation of this loss for parameter optimization, we embed an property predictor within the decoder. This auxiliary network predicts target properties from decoded or generated sequence representations, with prediction errors directly contributing to the preservation loss. Formally, the total loss function is defined as follows.

$$\mathcal{L}_{\text{Total}} = \beta \left[ -\frac{1}{2} \sum_{i=1}^{N} \left( 1 + \log \sigma_i^2 - \mu_i^2 - \sigma_i^2 \right) \right]$$
$$+ \sum_{i=1}^{N} \text{CrossEntropy}(x_i', x_i) \tag{1}$$
$$+ \sum_{i=1}^{N} \left\| \mathbf{p}_i' - \mathbf{p}_i \right\|_2^2$$

In the right-hand side of Eq 1, the first item refers to the regularization term for latent variable distribution, equivalent to the KL divergence in VAE. In this formula, $\mu_i$ denotes the mean, and $\sigma_i^2$ represents the variance of the latent space. The second item refers to the cross entropy between the reconstructed or generated AMP (i.e., $x_i'$) and the input AMP (i.e. $x_i$). The third item refers to the property preserving loss, aiming to minimize the gap between property values (i.e., $\mathbf{p}_i'$) of the

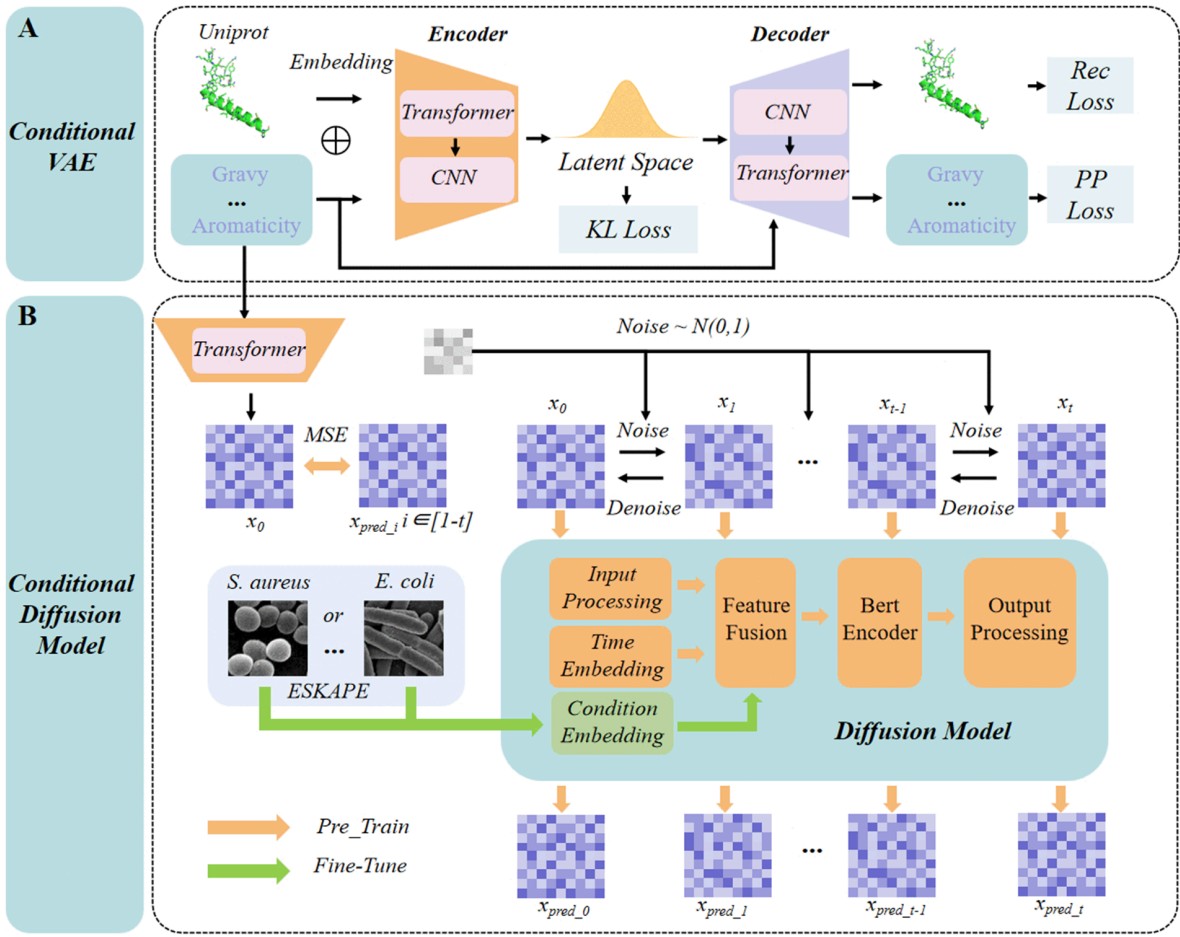

**Fig 2**. The detailed workflow for designing AMPs targeting the specific bacteria.

generated AMP and the targeted or expected property values (i.e., $\mathbf{p}_i$). It is important to note that we introduce a weighting coefficient $\beta$ in front of the KL divergence term to control its strength. If $\beta$ is too large at the beginning of training, the model is forced to match the prior distribution too strictly, which can lead to posterior collapse and prevent the encoder from learning informative latent representations [36]. To avoid this, we adopt an annealing strategy in which training starts with a small $\beta$, allowing the model to first focus on accurate reconstruction, and $\beta$ is then gradually increased to its target value. In contrast, the weights of the reconstruction loss and the property-preserving loss are kept fixed, since after normalization of the target properties these two terms operate on comparable scales.

## 2.3 Conditional diffusion model

**2.3.1 Model structure.** In the conditional diffusion model, we employ a BERT-Encoder instead of a simple Transformer structure to leverage its bidirectional contextual representation capabilities (Fig 2B). A more detailed description of the model is provided in S1 Appendix. This enables the model to better capture long-range dependencies and address the local convergence issues in diffusion models [37]. During pre-training, we freeze the condition embedding module, while during fine-tuning, we activate this module. We assign distinct labels to samples with MIC values below 10 $\mu$M and above 20 $\mu$M for contrastive training.

Specifically, in both the pre-training and conditional fine-tuning phases, the noise injection process follows Eq 2, where $x_0$ represents the original data and $\alpha_t$ denotes the noise scheduling parameter at diffusion step $t$. In the diffusion model, $\bar{\alpha}_t = \Pi_{i=1}^t \alpha_i$ represents the cumulative product of the noise scheduling parameters up to step $t$. This cumulative product determines the overall noise level at each diffusion step, affecting the final denoised output. The use of $\bar{\alpha}_t$ allows the model to maintain a consistent noise schedule throughout the diffusion process, enabling more stable and controlled generation.

$$
\begin{aligned}
x_t &= \sqrt{\alpha_t}x_{t-1} + \sqrt{1-\alpha_t}\epsilon_{t-1} \\
&= \sqrt{\alpha_t\alpha_{t-1}\cdots\alpha_1}x_0 + \sqrt{1-\alpha_t\alpha_{t-1}\cdots\alpha_1}\epsilon \\
&= \sqrt{\bar{\alpha}_t}x_0 + \sqrt{1-\bar{\alpha}_t}\epsilon
\end{aligned}
\tag{2}
$$

The distinction between the two phases lies in the denoising procedure: during pre-training, conditional embedding remains frozen, and the predicted original data is derived through Eq 3. In contrast, during fine-tuning, this conditional mechanism is activated, leading to the modified formulation in Eq 4. Here, data samples with high antibacterial activity (MIC values below 10 $\mu$M) are assigned label 0 while those with low activity (MIC values above 20 $\mu$M) receive label 1. These categorical labels are subsequently embedded to obtain the conditional vector $c$. Additionally, in the equations, $t$ represents the current time step, , and $f_\theta$ denotes the parameterized model used to predict the data.

$$
x_{pred\_t} = f_\theta(x_t, t) \tag{3}
$$
$$
x_{pred\_t} = f_\theta(x_t, t, c) \tag{4}
$$

**2.3.2 Loss function.** In the implementation of conditional diffusion model, the mean square error is used as the loss function (as shown in Eq 5), aiming to directly predict the real data during the training process. Note that the direct alignment of the predicted values at each time step can explicitly constrain the error propagation path in the Markov chain and effectively alleviate the error accumulation problem caused by noise predictions [38].

$$
\mathcal{L}_{\text{Total}} = \frac{1}{t}\sum_{i=1}^{t}\left\|x_i^{start} - x_{pred\_i}\right\|_2^2 \tag{5}
$$

In the above loss definition formula, $t$ represents the total number of timesteps in the diffusion model; $x_i^{start}$ is the linear combination of the initial embedding representation $x_0$ and the noise; $x_{pred\_i}$ refers to the predicted representation generated by the diffusion model at the $i$-th timestep, which reflects the model's approximation of the target at the current timestep. By calculating the squared Euclidean distance between these two representations, the discrepancy between the predicted values and the ground truth is quantified.

## 2.4 Minimum inhibitory concentration predictor

**2.4.1 Model structure.** Through training on AMP sequences and bacterial genomic sequences, the predictor is implemented by a multi-layer cascade framework to predict the MIC of AMPs against specific target pathogens (Fig 3). First, AMP sequences and their corresponding MIC values for the target pathogen (e.g., *E. coli* or *S. aureus*) are extracted from the GRAMPA database. The bacterial genomic sequences are then converted into numerical feature vectors via *k*-mer encoding (*k*=6 in this study). Subsequently, an attention mechanism is employed to analyze functional associations between sequence fragments. Finally, a MLP (Multilayer Perceptron) is used to derive the predicted MIC value of the AMP against the target pathogen. This framework enables training of pathogen-specific MIC predictors by independently configuring the MIC training dataset and the genomic sequence of the target bacterium. In our study, the MIC predictor plays distinct roles during the generation and evaluation phases. During the generation phase, the MIC predictor is

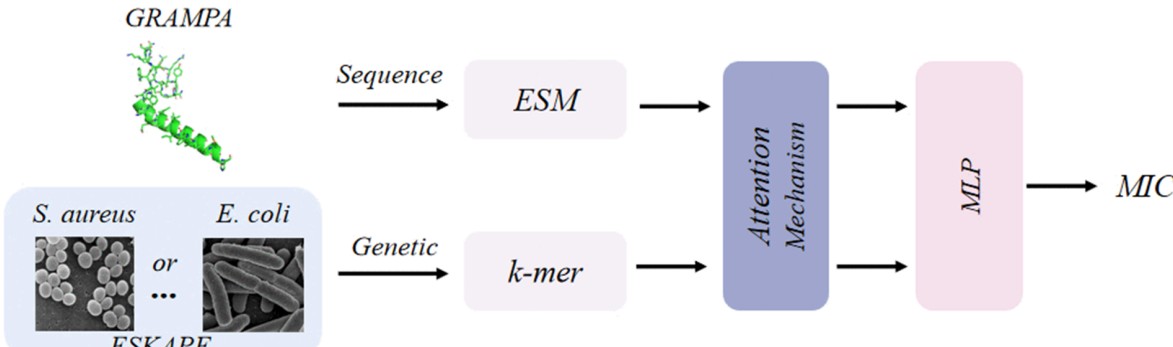

**Fig 3. The overall architecture of MIC predictor.** Note that ESM (Evolutionary Scale Modeling) [39] is a pretrained protein language model for sequence embedding, while MLP (Multilayer Perceptron) is a feed-forward neural network for prediction.

not directly involved in generating AMPs; instead, it is used in the evaluation phase to assess the predicted antimicrobial activity of the generated peptides against specific bacteria.

**2.4.2 Loss function.** For MIC predictor, we directly use MSE as the loss function, which is presented in Eq 6.

$$\mathcal{L}_{\text{MIC}} = \frac{1}{N} \sum_{i=1}^{N} \left( \text{MIC}_{\text{true}}^{(i)} - \text{MIC}_{\text{pred}}^{(i)} \right)^2 \tag{6}$$

More specifically, in Eq 6, $\text{MIC}_{\text{true}}^{(i)}$ represents the ground-truth MIC value of the $i$-th sample, $\text{MIC}_{\text{pred}}^{(i)}$ denotes the predicted MIC value for the $i$-th sample and $N$ denotes the specific sequence number in the dataset.

## 2.5 Training phase

The training phase includes four distinct steps: pre-training the CVAE, pre-training the diffusion model, fine-tuning the conditional diffusion model, and training MIC predictors. Specifically, the pre-training of CVAE and diffusion model serves as essential prerequisites for subsequent AMP generation fine-tuned for specific bacterial targets. During the fine-tuning step, both pre-trained models are loaded to initialize the framework. Subsequently, supervised discriminative fine-tuning is performed on the conditional embedding layer of the diffusion model using MIC-stratified datasets, enabling AMP generation for treating targeted pathogens. This process yields models with distinct parameters for different bacterial targeted pathogens, thereby optimizing antimicrobial specificity. Finally, specialized predictors are trained using MIC data and genomic sequences specific to individual bacterial targets to support experimental validation. Notably, these predictors can operate independently, providing researchers with an efficient tool for rapid prediction of AMP candidate efficacy in downstream applications.

## 2.6 Generating phase

Our framework enables the generation of AMPs with tailored property values against specific target bacteria. The pseudo codes of the generation process are summarized in Algorithm 1.

The algorithm initiates from a random noise $x_t$ and progressively transforms it through a $T$-step denoising process into a latent representation exhibiting low Minimum Inhibitory Concentration and high antibacterial activity. This refined latent state then undergoes reparameterization in the latent space to derive the probabilistic latent variable $z$. Subsequently, $z$ is concatenated with editable property information (i.e., the designed physicochemical properties) and fed into a decoder to generate antimicrobial peptide sequences that satisfy predefined antibacterial efficacy and physicochemical properties.

## Algorithm 1 Antimicrobial peptide generation.

1: **Input:** Time steps: $\boldsymbol{T}$, noise: $\epsilon \sim \mathcal{N}(0, I)$, targeted bacteria: $c$, targeted property: $\boldsymbol{p}$

2: **Notation:** Signal retention at step $\boldsymbol{t}$: $\alpha_t$; a function parameterized by $\theta$ that maps $x_0$ to the mean: $f_\theta^{\text{mean}}$; a function parameterized by $\phi$ that maps $x_0$ to the logarithm of the variance: $g_\phi^{\text{logvar}}$

3: **Output:** Generated AMP

4: **A: Initialize noise**

5: $x_T \sim \mathcal{N}(0, I)$

6: **B: Diffusion Process (Denoising)**

7: **for** $t = T$ **downto** 1 **do**

8:     $x_{\text{pred\_t}} = f_\theta(x_t, t, c)$

9:     $\beta_t = 1 - \alpha_t, \quad \bar{\alpha}_t = \Pi_{i=1}^t \alpha_i$

10:    $\sigma_t = 1 - \frac{\bar{\alpha}_{t-1}}{1 - \bar{\alpha}_t} \beta_t$

11:    $x_{t-1} = \frac{\sqrt{\bar{\alpha}_{t-1}}\beta_t}{1 - \bar{\alpha}_t} x_{\text{pred\_t}} + \frac{\sqrt{\alpha_t}(1 - \bar{\alpha}_{t-1})}{1 - \bar{\alpha}_t} x_t + \sigma_t \epsilon$

12: **C: Sample Process**

13: $\mu = f_\theta^{\text{mean}}(x_0)$

14: $\log \sigma^2 = g_\phi^{\text{logvar}}(x_0)$

15: $z = \mu + \epsilon \cdot \exp(0.5 \cdot \log \sigma^2)$

16: **D: Decoder Procedure**

17: $\textbf{AMP} = \textbf{Decoder}(\textbf{z} \parallel \textbf{p})$

# 3 Results

## 3.1 Implementation

In this study, the proposed framework is implemented on Pytorch. The detailed configuration of hyperparameters is listed as follows: the learning rate for CVAE is set to 7e-4 with a batch_size of 512; the diffusion model employs a learning rate of 1e-4, a batch_size of 512, and timesteps of 500; the MIC predictor utilizes a learning rate of 1e-4, incorporates the ESM-1b model, and sets $k$=6 for $k$-mer analysis. Moreover, to test our framework, we trained it on two of the most common targeted bacteria, i.e., *E. coli* and *S. aureus*.

Generally, the whole training process can be completed efficiently, and the effective and robust generative models are derived accordingly. Due to the limit of space, the detailed presentation of the training process is provided in S4 Appendix. Note that the whole implementation can be completed on the RTX 3090 (24G) graphics card. Specifically, the CVAE model was trained with a batch size of 512 sequences per step using a 3-layer Transformer encoder/decoder, and a full 50-epoch run completed within one day. The diffusion model was trained under the same setup; 200 epochs required only ten hours, and fine-tuning the denoiser for a new bacterial species required less than half an hour, demonstrating that adaptation to additional bacteria can be achieved efficiently. The MIC predictor, which is a lighter regression model trained separately for *E. coli* and *S. aureus*, could be trained end-to-end in under one hour per organism.

## 3.2 Beyond existing approaches: Quantifying the advantages of our model

To validate the performance of our model in generating AMPs targeting specific bacteria, we compared it with five representative models in the field of de novo AMPs design, especially focusing on the newly released models in 2025: LLM-based [29], Latent Diffusion [28], and PPGC-DVAE [26]. For the comparison metrics, we selected the minimum inhibitory concentration (MIC) of AMPs generated by these models targeting two bacteria (*E. coli* and *S. aureus*). To ensure a fair comparison, the pre-trained MIC predictor tailored for different targeted bacteria was employed for all models, and each model generated 512 sequences.

For *E. coli* (Fig 4A) and *S. aureus* (Fig 4B), our model (dark red) consistently shows significantly lower median MIC values of generated AMPs compared to the other baseline models, indicating stronger antimicrobial effects of the generated

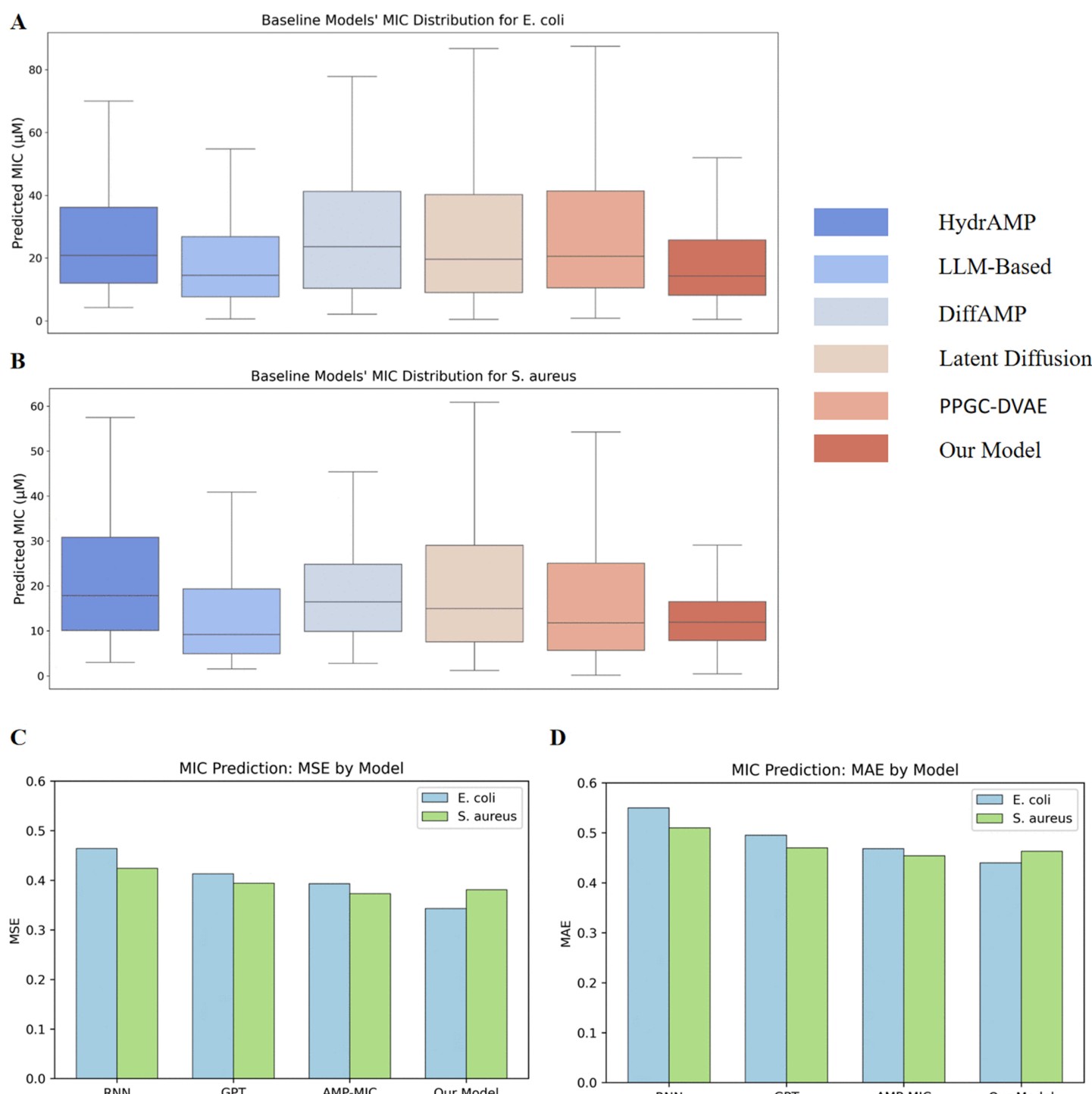

**Fig 4. Results of performance comparison with baseline models.** (A) Predicted MIC for *E. coli* of AMPs generated by six different models. (B) Predicted MIC for *S. aureus* of AMPs generated by six different models. (C) MSE values for different MIC predictors. (D) MAE values for different MIC predictors.

AMPs. Based on the predicted MIC for *S. aureus*, generated AMPs by our model also demonstrate lower and more stable MIC values, reflecting superior antibacterial activity and stability across different bacteria. Specifically, for *S. aureus*, generated AMPs by our model not only exhibit a lower MIC but also show a more concentrated distribution of predicted MIC values, indicating the higher stability. In contrast, AMPs derived by baseline models like Latent Diffusion and PPGC-DVAE show higher MIC values with a wider distribution, suggesting generated AMPs with greater variability of predicted MIC values. Therefore, our model demonstrates stronger antimicrobial effects and stability in AMP generation, proving its superiority in this field of de novo AMP design. Detailed comparison results are presented in Table 1.

In addition, we compared our MIC predictor with three representative MIC predictors on both *E. coli* and *S. aureus*, using mean squared error (MSE) and mean absolute error (MAE) as evaluation metrics. The results show that our model performs better than or comparable to the baseline models. Notably, for predicting the MIC of AMPs against *E. coli*, our model achieves the best MSE and MAE. Although on *S. aureus* our metrics are slightly lower than those of AMP-MIC, the difference in MSE is only 0.008, and our model already shows a substantial improvement over the RNN-based and GPT-based baselines. This further confirms the effectiveness of our MIC predictor. Detailed comparison results are presented in Table 2.

This above analysis highlights that not only in terms of generating AMPs with strong antimicrobial effects, but also in predicting their activity with greater accuracy, the proposed model outperforms baseline models.

## 3.3 Novelty and diversity analysis of generated AMPs

To validate that our generative model can produce novel and diverse peptides, we also evaluated the novelty and diversity of sequences generated by each model (Table 3). For all models, the data used to evaluate novelty and diversity are consistent with those in Sect 3.2 (i.e., each model generates 512 sequences). To quantify the *n*-gram-based novelty of the generated AMPs, we constructed a reference set of *n*-gram from all experimentally validated AMPs in the GRAMPA database, which provides a high-quality, curated collection of AMPs. All sequences were first merged into a single FASTA file, and we then extracted all contiguous 4-gram and 6-gram using a sliding window with stride 1. The union of these *n*-gram defines the reference set $N_{ref}$. For each set of generated peptides, we likewise extracted all contiguous 4-gram or

**Table 1**. Results of performance comparison on MIC ratio metrics.[a]

| Model | Target E. coli ($\mu$M) | | | Target S. aureus ($\mu$M) | | |
|---|---|---|---|---|---|---|
| | 0-20 | 20-40 | 40+ | 0-20 | 20-40 | 40+ |
| HydrAMP | 0.4819 | 0.3253 | 0.1928 | 0.5444 | 0.2666 | 0.1890 |
| LLM-based | 0.6252 | 0.2505 | 0.1243 | 0.7559 | 0.1614 | 0.0827 |
| DiffAMP | 0.4418 | 0.2790 | 0.2792 | 0.6315 | 0.3233 | 0.0452 |
| Latent Diffusion | 0.5032 | 0.2429 | 0.2539 | 0.6271 | 0.2139 | 0.1590 |
| PPGC-DVAE | 0.4930 | 0.2465 | 0.2605 | 0.6736 | 0.2130 | 0.1134 |
| Our Model | 0.6508 | 0.2374 | 0.1118 | 0.8500 | 0.1301 | 0.0199 |

[a]The evaluation values in the table represent the proportion of AMPs that meet the specified criteria.

**Table 2**. Comparative evaluation for different MIC predictors.

| Model | Target E. coli | | Target S. aureus | |
|---|---|---|---|---|
| | MSE ↓ | MAE ↓ | MSE ↓ | MAE ↓ |
| RNN | 0.464 | 0.550 | 0.424 | 0.510 |
| GPT | 0.413 | 0.495 | 0.394 | 0.470 |
| AMP-MIC | 0.393 | 0.468 | **0.373** | **0.454** |
| Our Model | **0.343** | **0.440** | 0.381 | 0.463 |

PLOS Computational Biology

**Table 3**. Comparison results on novelty and diversity of sequences.

| Model | Novelty (6-gram) ↑ | Novelty (4-gram) ↑ | Diversity (Entropy) ↑ | Diversity (Levenshtein) ↑ |
|---|---|---|---|---|
| HydrAMP | 0.9532 | 0.7131 | 2.8071 | 5.0332 |
| LLM-based | 0.9792 | 0.7148 | 2.8624 | 17.4223 |
| DiffAMP | 0.9971 | 0.7783 | 3.0696 | 14.8454 |
| Latent Diffusion | 0.9935 | 0.8276 | 2.9773 | 22.0056 |
| PPGC-DVAE | 0.9990 | 0.9083 | 2.8366 | **22.8948** |
| Our Model | **0.9998** | **0.9484** | **3.3115** | 20.3477 |

6-gram with stride 1 and collected the unique *n*-grams into a set $N_{new}$. The novelty score is defined as follows.

$$\text{Novelty} = 1 - \frac{|N_{new} \cap N_{ref}|}{|N_{new}|}. \tag{7}$$

In addition to novelty, we also introduce diversity analysis based on Shannon entropy and Levenshtein distance to evaluate the variability of the generated peptides [40,41]. Shannon entropy is a standard method for quantifying sequence diversity, and it allows us to assess how varied the amino acid distribution is across different peptides. Specifically, we calculate the Shannon entropy for each generated peptide sequence *X* using the following formula:

$$H(X) = -\sum_{i=1}^{n} p(x_i) \log_2 p(x_i) \tag{8}$$

where $p(x_i)$ represents the probability of amino acid $x_i$ occurring at a particular position in the sequence, and *n* denotes the number of unique amino acids in the sequence. A higher entropy value indicates a more diverse amino acid composition. Finally, we obtain the average entropy value of the sequences generated by each model as the diversity metric. On the other hand, the diversity value based on Levenshtein distance is calculated by determining the minimum number of edit operations (including insertion, deletion, and substitution) required to transform one sequence into another, which effectively quantifies the similarity between two sequences. Specifically, for the sequences generated by each model, we compute the average Levenshtein distance between all pairs as the final diversity value.

The results show that our model exhibits a high level of novelty in generating AMPs, achieving 6-gram and 4-gram scores of 0.9998 and 0.9484, respectively. Compared to other models, our approach demonstrates a clear advantage in terms of novelty, especially when compared to models such as HydrAMP, LLM-based, and DiffAMP, which show significantly lower overlap, indicating that our model generates more unique AMPs. In addition, compared to other models, our model achieved the highest diversity score (3.3115) according to Shannon entropy, indicating that the peptides it generates have the greatest variability in amino acid distribution. Although our model did not achieve the highest Levenshtein distance score, it derives the comparable performance with the top-2 best models (i.e., "Latent Diffusion" and "PPGC-DVAE"). This further emphasizes our model's ability to generate peptides with significant diversity and provides additional evidence of its robustness in generating a wide variety of AMPs.

### 3.4 Evaluation of ability for controlling physicochemical properties of generated AMPs

To verify that our proposed framework can generate AMPs with programmable properties for treating targeted bacteria, we conducted comprehensive experiments. In the subsequent experiments, 512 sequences were generated for each target property. First, we compared the distribution of ten properties between two sets of AMPs generated with the average properties (shown in S5 Appendix) for *E. coli* and *S. aureus* (Fig 5A) and the AMPs generated under unconditioned conditions (Fig 5B). It is clearly observed that the AMPs generated by our framework exhibit clustering of attributes around the

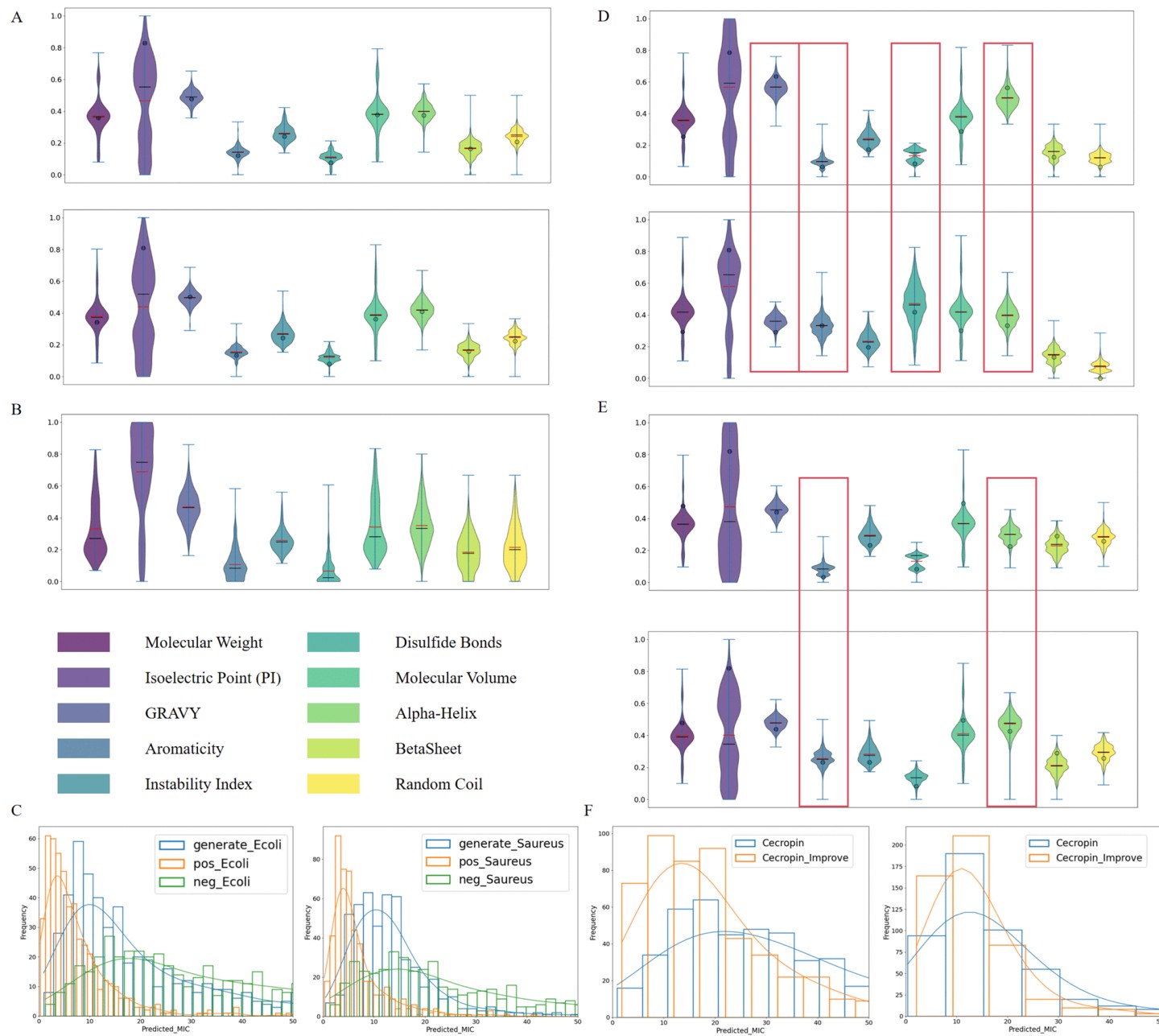

**Fig 5. Evaluation of the conditional generative ability.** (A) Conditional Generation. The *y*-axis represents normalized property values. (B) Unconditional Generation. (C) Comparison of MIC distributions. This panel compares the predicted MIC distributions of model-generated peptides with those of the positive and negative samples from the training set. (D) Comparison of the resulting physicochemical properties of generated peptide sequences under two different target property specifications. The red boxes highlight regions where differences exist between the two property groups. (E) Adjusting the properties of *Cecropin* and comparing the distributional differences in the generated peptides. (F) Enhancement of MIC for *Cecropin*. The left panel shows results against *E. coli*, and the right panel shows results against *S. aureus*.

targeted or expected values in almost all properties, with the exception of Isoelectric Point. However, the property distributions of the unconditioned AMPs are all quite dispersed as shown in Fig 5B. Interestingly, the isoelectric point (pI) distribution remains fairly even. The pI of a peptide is primarily determined by the balance between basic, positively charged

residues and acidic, negatively charged residues. For short peptides, even small changes in the number of these residues can cause the pI to shift abruptly into a highly cationic range, whereas the introduction of a few acidic residues can lower the pI toward more neutral values. Therefore, the pI tends to show a more balanced distribution over a broader range rather than being concentrated at a single high-pI value. This is consistent with the established view that AMPs are generally cationic, but their activity arises from an overall balance of charge rather than from unrestricted maximization of positive charge [42].

In addition, for each type of pathogen (i.e., *E. coli* and *S. aureus*), we compared the distribution of predicted MIC values of generated AMPs with the distribution of predicted MIC values of positive and negative samples used in fine-tuning diffusion model (Fig 5C). It can be clearly observed that the distribution of generated AMPs is between those of positive and negative samples, indicating that we can generate AMPs with high antibacterial activity for treating targeted types of bacteria. Moreover, we conducted a detailed analysis of the length distribution and amino acid composition of these two groups of peptides, as described in S2 Appendix.

Next, we analyzed the AMPs generated with different input conditions (i.e., different expected physicochemical properties). Here, we extracted two sets of properties from the *E. coli*-targeted AMPs, which are different significantly in GRAVY, Aromaticity, Disulfide Bonds, and $\alpha$-helix, and showing little difference in other properties (The details of two property sets are provided in S5 Appendix). The final generated two sets of AMPs (Fig 5D) exhibits differences in the distributions of the significantly differing properties, while showing nearly identical distributions in the properties with little difference. This observation further confirms that our model is able to learn the input properties, effectively generating AMPs with programmable or controllable physicochemical properties.

Subsequently, we focused on improving the properties of an AMP obtained from nature to observe whether enhancing certain properties would increase the antimicrobial activity. In this study, we selected *Cecropin*, a relatively short linear $\alpha$-helix cationic peptide that shows solubility against Gram-negative and Gram-positive bacteria, as well as filamentous fungi, yeast, and some protozoa [43]. In fact, insect-derived *Cecropin* and *Cecropin*-like compounds have been shown to inhibit multiple drug-resistant bacteria and even suppress the proliferation of Human Immunodeficiency Virus 1 (HIV) [44]. Existing research has indicated that increasing the proportion of aromatic amino acids and $\alpha$-helix content can enhance its antimicrobial activity [44]. Biologically, $\alpha$-helix structures are known to enhance membrane disruption, as they promote hydrophobic interactions with bacterial membranes, facilitating membrane penetration [45]. Aromatic residues, due to their hydrophobic nature, further enhance the peptide's ability to bind to and disrupt lipid membranes [46]. Together, these features increase the effectiveness of AMPs by promoting their interaction with bacterial membranes, leading to more potent antimicrobial activity. Therefore, we increased the values of these two properties by 0.2 based on the real properties of *Cecropin* (shown in S5 Appendix), and the final generated AMP showed a clear difference in the attribute distribution of these two properties (Fig 5E). This observation suggests that our model has the potential to optimize existing (suboptimal or undesirable) AMPs by editing the corresponding physicochemical properties.

Furthermore, we performed MIC predictions for two sets of AMPs targeting *E. coli* and *S. aureus*, and the results showed improvement of antimicrobial activity (Fig 5F). Specifically, for *E. coli*, the proportion of AMPs with MIC values below 20 $\mu$M increased from 0.3262 to 0.6348; and for *S. aureus*, the proportion increased from 0.6934 to 0.8223. In addition, we used ToxinPred 3.0 [47] to analyze the generated sequences and found that the proportion of Non-Toxin increased from 0.9766 to 0.9941 (Table 4). The above analysis show that our model has its advantages in verifying the relationship between certain properties and the antimicrobial activity of AMPs.

## 3.5 Visualization analysis of generated AMPs' hidden representation

To better validate the effectiveness of the proposed framework, we conducted a visualization analysis by comparing the embedded representations of positive and negative samples during fine-tuning process. More specifically, the hidden representations of positive and negative samples are obtained from the intermediate layers during generation, and are

**Table 4.** Comparison of evaluation indicators between Cecropin and Cecropin-Improve.[a]

|  | *Cecropin**[*] | *Cecropin_Improve*[*] |
|---|---|---|
| MIC_Target *E. coli* (< **20**$\mu$M) | 0.3262 | **0.6348** |
| MIC_Target *S. aureus* (< **20**$\mu$M) | 0.6934 | **0.8223** |
| Non-Toxic (Threshold 0.5) | 0.9766 | **0.9941** |

[a]The values in the table represent the proportion of AMPs that meet the specified criteria.

[*]The first column corresponds to AMPs generated using the properties of *Cecropin* as the target properties.

[*]The second column corresponds to AMPs generated using the properties of the improved *Cecropin* as the target properties.

visualized based on t-SNE embedding [48]. As shown in Fig 6A, the embedded representations of positive samples (MIC <10 $\mu$M) and negative samples (MIC >20 $\mu$M) are roughly separated, indicating that the embedding module has successfully captured the intrinsic differences between the two categories of samples. This enables more efficient and stable fine-tuning of the conditional diffusion model.

After fine-tuning with this set of positive and negative samples, we used two different conditions (targeted vs. non-targeted) as inputs to the diffusion model to obtain two sets of sampling distributions from the latent space. These were visualized using contour plots (Fig 6B) and t-SNE [48] (Fig 6C). Based on these three figures, we observe that under the targeted condition, the sampling distribution in the latent space shows concentrated and high-density hidden features, suggesting that the representations of high antimicrobial activity AMPs are consistent in the latent space.

## 3.6 Discovery of novel AMPs

To validate the capability of our model in generating novel and potent antimicrobial peptides (AMPs), we tested it against two pathogenic bacteria, i.e., *E. coli* and *S. aureus*. For each targeted bacterium, we selected the top 10 AMP candidates (detailed information is provided in Table 5) exhibiting high antimicrobial activity during comprehensive simulation tests including hemolytic activity, cytotoxicity, and membrane interaction mechanisms. Subsequently, through rigorous screening, we identified four star candidate AMPs. Note that in the following analysis, the two AMPs targeting *E. coli* are denoted as Seq1-7 and Seq1-9, while the two AMPs targeting *S. aureus* are denoted as Seq2-5 and Seq2-10.

Specifically, we generated 100,000 AMP sequences for each target bacterial species. These sequences were analyzed and filtered using an MIC predictor and AMPScanner v2.0 [49], retaining the top 10 AMPs for subsequent characterization. Hemolytic potential was evaluated through the regression module of HemoPI2 [50], which predicts hazardous concentrations (HC50) or half-maximal effective concentrations (EC50) in micromolar units ($\mu$M). This factor represents the peptide concentration required to induce 50% lysis of red blood cells (RBCs). Toxicity assessment was performed using ToxinPred3.0, where lower Hybrid Score values correspond to higher probabilities of peptide sequence non-toxicity. Lastly, we performed molecular docking experiments using the HDOCK server [51]. We adopted a standard blind docking protocol with default settings. Specifically, no binding site constraints were predefined (i.e., an exhaustive search was carried out over the entire receptor surface); instead, each candidate peptide was docked to the target, and the top-ranked poses were selected based on the server's default scoring function. The docking score is a key metric in this analysis and was calculated using the iterative scoring functions ITScorePP or ITScorePR. A less negative docking score indicates a higher likelihood of binding, with a score around -200 or lower being ideal. A confidence score greater than 0.7 suggests a high probability of binding between the two molecules, a score between 0.5 and 0.7 indicates a possible binding interaction, and a score below 0.5 suggests a low likelihood of interaction. We used CHARMM-GUI [52] to construct rectangular lipid bilayers to simulate the cell membranes of *E. coli* and *S. aureus*. Although the membrane compositions of these two bacteria are different, both contain phosphatidylglycerol (POPG) and phosphatidylethanolamine (POPE). For the *E. coli* membrane, we set the POPG:POPE ratio to 2:8, while for the *S. aureus* membrane, the ratio was set to 8:2 [53]. These

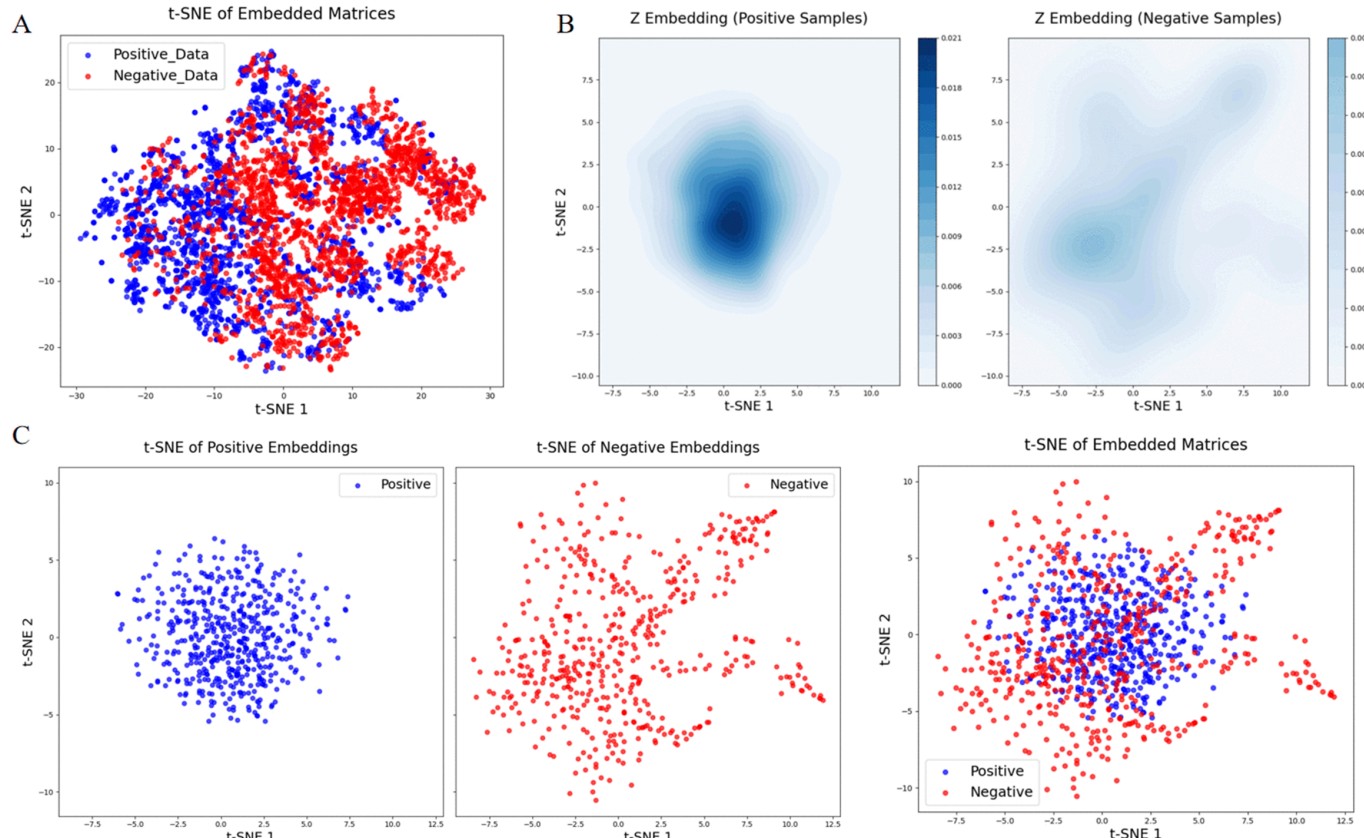

**Fig 6**. **Visualization of hidden representations.** (A) Visualization of hidden representations after Encoder of CVAE for positive data and negative data. (B) Contour plots to demonstrate generated positive samples and negative samples. (C) Visualization of t-SNE embedding for generated positive samples and negative samples (n_components = 2, perplexity = 30, n_iter = 1000, random_state = 43).

**Table 5**. **Top 20 AMPs for targeting *E.coli* and *S.aureus*.**

| Sequence ID | Targeted Bacteria | Sequence |
|---|---|---|
| Seq1-1 | E. coli | NRNKYLPLHQYNYSYQRTYFGLSM |
| Seq1-2 | E. coli | GGYMPNPYQYNVKYRLFP |
| Seq1-3 | E. coli | GAKYLMYLRSNFLYYGSR |
| Seq1-4 | E. coli | GKFSLLGKYFKELYSAFKKLNNKWS |
| Seq1-5 | E. coli | GRNYSMRCWQKMYSRLSY |
| Seq1-6 | E. coli | RKFSTYWQYYLRLPLS |
| Seq1-7 | E. coli | SRNKYQMYHMSYFRLL |
| Seq1-8 | E. coli | RKWKYMYEKQISKPLPQPLFS |
| Seq1-9 | E. coli | ALVAKLKLVAKSIKYNPNPQYYGYL |
| Seq1-10 | E. coli | AGKYFLSKRYLNLYK |
| Seq2-1 | S. aureus | RFWNPLWFRLRER |
| Seq2-2 | S. aureus | GWMHWRPLFL |
| Seq2-3 | S. aureus | NPMRCPFRLLKWY |
| Seq2-4 | S. aureus | GPMPSFRYWRYYPLRIL |
| Seq2-5 | S. aureus | PRWPPYPERYPFKYMRLRLLL |
| Seq2-6 | S. aureus | GPWLRLWFRFS |
| Seq2-7 | S. aureus | RPRPLRPWYNFFNLQRLLTAM |
| Seq2-8 | S. aureus | ERNPSFFWKRINYTLYKLYRALSG |
| Seq2-9 | S. aureus | PLPYYWFYKLLPRLRETG |
| Seq2-10 | S. aureus | RPWYLRPFFR |

membranes were embedded in an aqueous solution containing K$^+$ and Cl$^-$ ions. For all AMPs, their three-dimensional structures were predicted using AlphaFold2 [54].

The final simulation results are presented in Fig 7. These four star candidate AMPs exhibit superior hemolytic and toxicological profiles, with hemolytic activities exceeding the threshold of 100 $\mu$M (HC50 >100 $\mu$M) and near-zero toxicity values. They further demonstrate favorable molecular docking scores (<-200) and achieve confidence scores above 0.85. These observations conclusively demonstrate their strong potential for clinical translation and application. The simulation analysis data for all candidate AMPs are shown in Tables 6 and 7. Due to the limit of space, the docking analysis of the remaining candidate peptides with the cell membrane is shown as well in S6 Appendix.

## 4 Discussion

The physicochemical properties of antimicrobial peptides (AMPs) critically influence their membrane-disruptive capability, which in turn governs their antimicrobial efficacy. Consequently, targeted modulation of specific physicochemical properties presents a strategic approach for the rational design of AMPs with enhanced antimicrobial potency. In this work, we propose a novel framework for designing/generating AMPs that are targeted toward specific bacteria, with editable physicochemical properties. To assist in the analysis and design of these AMPs, we have also developed a corresponding MIC Predictor to predict their antimicrobial activity. Unlike previous approaches that simply design a series of broad-spectrum AMPs for analysis, our framework allows for fine-tuning a generative model specifically targeted toward a particular bacterial strain. This enables more efficient acquisition of AMPs with high antimicrobial activity and target specificity.

To validate the effectiveness of our framework, we implemented two generative models targeting *E. coli* and *S. aureus* as representatives of Gram-negative and Gram-positive bacteria, respectively. Through extensive simulation experiments, we demonstrated that our models can not only generate AMPs with targeted specificity but also allow for adjustments of specific physicochemical properties to achieve enhanced antimicrobial efficacy. Beyond designing highly potential AMPs, our framework possesses an inherent advantage in enabling researchers to experimentally validate potential correlations between critical physicochemical properties and multifaceted AMP performance metrics, including antimicrobial efficacy, cytotoxicity, and hemolytic activity. This systematic investigation was previously constrained by the scarcity of diverse AMP data, whereas our platform facilitates high-throughput generation of AMP variants with precisely controlled physicochemical profiles for comprehensive structure-activity relationship studies. To demonstrate this capability, we conducted a focused investigation on *Cecropin* derivatives by selectively modulating aromatic residue ratios and $\alpha$-helix propensity while maintaining other physicochemical properties constant. The resulting AMPs exhibited statistically significant enhancement in antimicrobial potency, experimentally confirming the critical role of these two properties in bactericidal activity.

In addition, we visualized the latent space. Positive sampling from this concentrated region allows the generative model to stably generate more high-antimicrobial-activity AMPs with similar properties and functions. In contrast, the distribution of negative samples in the latent space exhibits greater dispersion. This suggests that the latent representation of negative samples is more complex, potentially related to diverse noise patterns or variations in features. Compared to the high-density clustering of positive samples, the more dispersed nature of negative samples presents a higher challenge for the generative model when treating AMPs with low antimicrobial activity. This difference further highlights the generative model's focus on the generation direction of positive samples, optimizing its ability to generate AMPs with high-antimicrobial activity.

Although our framework is more advanced than existing de novo AMP generation models, it may still face a potential limitation: the efficacy against certain bacteria requires further validation, such as with *P. multocida*, *S. enteritidis*, and other bacteria beyond the ESKAPE pathogens. The training data for generating AMPs targeting these bacteria constitutes only 2.5% of the data available for *E. coli* and *S. aureus*. The scarcity of training data might lead to suboptimal performance in generating AMPs against these bacterial strains, which remains a challenge that we aim to address in

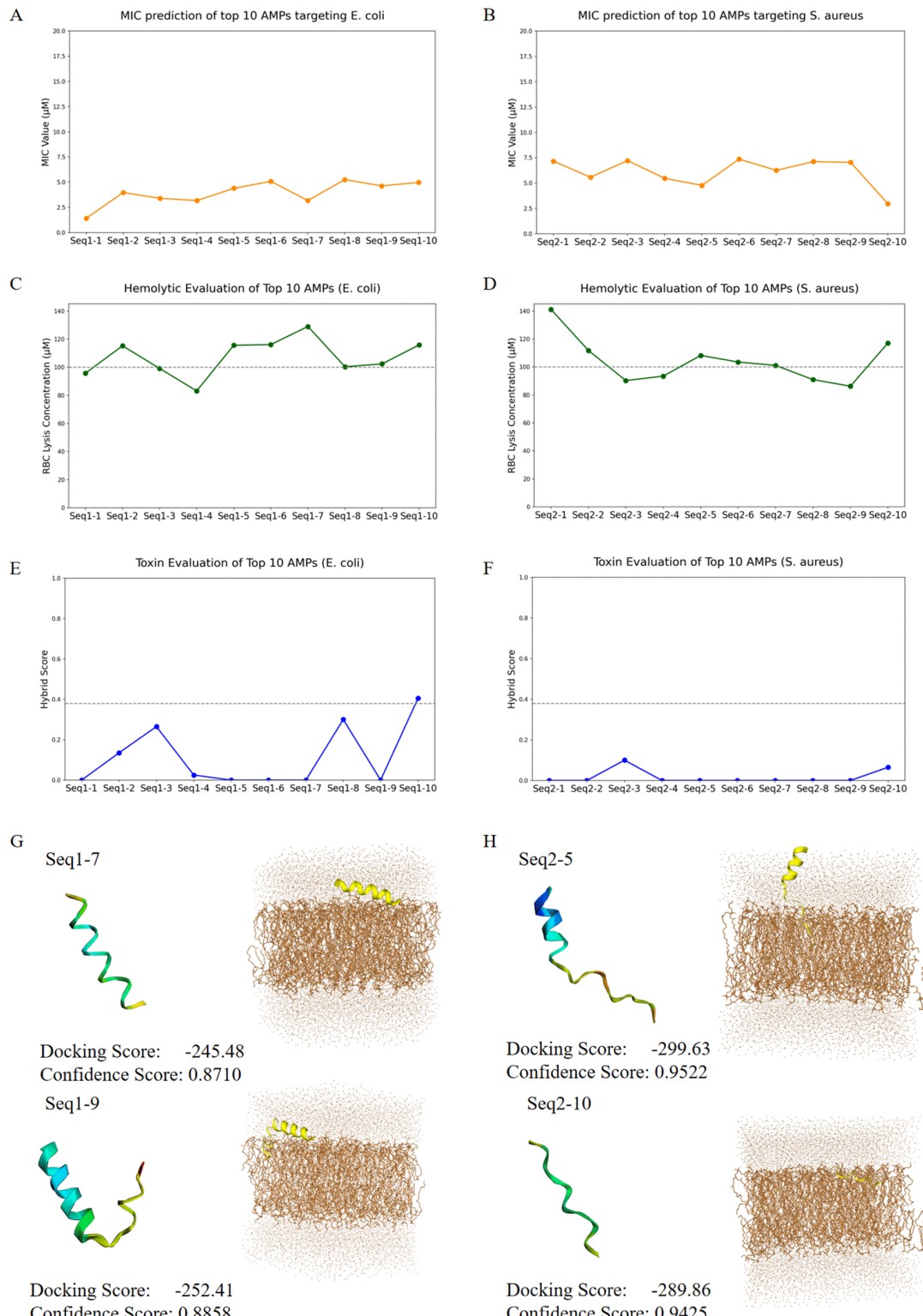

**Fig 7. In silico test of biological properties for identified AMPs.** (A) MIC Prediction Targeting *E. coli*. (A) MIC Prediction Targeting *S. aureus*. (C)(D) Hemolytic Evaluation of AMPs. (E)(F) Toxin Evaluation of AMPs. (G) Molecular Docking of AMP with *E. coli* Cell Membrane. (H) Molecular Docking of AMP with *S. aureus* Cell Membrane.

**Table 6**. Analysis of the antibacterial activity of the top 20 AMPs.

| Sequence ID | MIC (μM) ↓ | Docking Score ↓ | Confidence Score ↑ |
|---|---|---|---|
| Seq1-1 | 1.41 | −320.02 | 0.9677 |
| Seq1-2 | 3.98 | −259.89 | 0.9001 |
| Seq1-3 | 3.40 | −289.55 | 0.9422 |
| Seq1-4 | 3.18 | −256.92 | 0.8946 |
| Seq1-5 | 4.40 | −273.58 | 0.9211 |
| Seq1-6 | 5.08 | −250.66 | 0.8822 |
| Seq1-7 | 3.18 | −245.48 | 0.8710 |
| Seq1-8 | 5.26 | −276.99 | 0.9269 |
| Seq1-9 | 4.63 | −252.41 | 0.8858 |
| Seq1-10 | 4.99 | −231.27 | 0.8355 |
| Seq2-1 | 7.16 | −233.11 | 0.8405 |
| Seq2-2 | 5.57 | −242.17 | 0.8634 |
| Seq2-3 | 7.22 | −281.64 | 0.9329 |
| Seq2-4 | 5.47 | −244.54 | 0.8688 |
| Seq2-5 | 4.77 | −299.63 | 0.9522 |
| Seq2-6 | 7.36 | −242.62 | 0.8644 |
| Seq2-7 | 6.25 | −317.17 | 0.9659 |
| Seq2-8 | 7.12 | −309.25 | 0.9603 |
| Seq2-9 | 7.04 | −257.89 | 0.8964 |
| Seq2-10 | 2.97 | −289.86 | 0.9425 |

**Table 7**. Analysis of the hemolytic and toxic properties of the top 20 AMPs.

| Sequence ID | HC50 (μM) ↑ | Hemo/ Non-Hemo | Hybrid Score ↓ | Toxin/ Non-Toxin |
|---|---|---|---|---|
| Seq1-1 | 95.732 | Hemolytic | 0.000 | Non-Toxin |
| Seq1-2 | 115.186 | Non-Hemolytic | 0.135 | Non-Toxin |
| Seq1-3 | 99.112 | Hemolytic | 0.265 | Non-Toxin |
| Seq1-4 | 83.112 | Hemolytic | 0.025 | Non-Toxin |
| Seq1-5 | 115.591 | Non-Hemolytic | 0.000 | Non-Toxin |
| Seq1-6 | 116.092 | Non-Hemolytic | 0.000 | Non-Toxin |
| Seq1-7 | 128.933 | Non-Hemolytic | 0.000 | Non-Toxin |
| Seq1-8 | 100.296 | Non-Hemolytic | 0.300 | Non-Toxin |
| Seq1-9 | 102.312 | Non-Hemolytic | 0.000 | Non-Toxin |
| Seq1-10 | 115.861 | Non-Hemolytic | 0.405 | Toxin |
| Seq2-1 | 141.116 | Non-Hemolytic | 0.000 | Non-Toxin |
| Seq2-2 | 111.746 | Non-Hemolytic | 0.000 | Non-Toxin |
| Seq2-3 | 90.208 | Hemolytic | 0.100 | Non-Toxin |
| Seq2-4 | 93.420 | Hemolytic | 0.000 | Non-Toxin |
| Seq2-5 | 108.208 | Non-Hemolytic | 0.000 | Non-Toxin |
| Seq2-6 | 103.468 | Non-Hemolytic | 0.000 | Non-Toxin |
| Seq2-7 | 101.089 | Non-Hemolytic | 0.000 | Non-Toxin |
| Seq2-8 | 90.903 | Hemolytic | 0.000 | Non-Toxin |
| Seq2-9 | 86.212 | Hemolytic | 0.000 | Non-Toxin |
| Seq2-10 | 117.077 | Non-Hemolytic | 0.065 | Non-Toxin |

future work. In addition, the evaluation of the generated AMPs in this study relies on computational predictions. Therefore, the reported computational scores should be interpreted as heuristic prioritization rather than definitive evidence of efficacy or safety. All candidate peptides proposed by our framework will require systematic experimental validation in future work, including in vitro MIC measurements, toxicity and hemolysis assays, and in vivo infection model experiments. Furthermore, our framework demonstrates strong scalability. By adjusting the maximum sequence length and retraining the model, it can be extended to generate longer peptides. Additionally, by modifying the embedding representation, the

model can be applied to non-peptide antimicrobial agents, such as those encoded by SMILES, thereby opening a feasible direction for future research.

Lastly, the issue of antibiotic resistance remains a critical problem. Our framework is already capable of generating AMPs with specific properties for different resistant bacteria, and the generated AMPs demonstrate high antimicrobial activity and low toxicity. This confirms that our framework can be used to generate AMP candidates suitable for clinical applications, helping to address the globally growing antibiotic resistance crisis.

## 5 Conclusions

Antimicrobial peptides (AMPs) have garnered increasing attention from both the academic and medical communities as a promising strategy to address the growing crisis of bacterial resistance. To enable the precise design of desired AMPs, this study proposes a novel framework capable of generating peptide sequences with finely controlled physicochemical properties and specific strain-targeting capabilities. By decoupling and regulating the learning of physicochemical properties and strain-specific conditions at different stages, the framework can de novo design AMPs with high antimicrobial activity. Moreover, this innovative approach facilitates a deeper understanding of the relationship between the physicochemical properties of AMPs and their diverse functional characteristics, thereby offering insights into their underlying mechanisms of action. Ultimately, we hope this study will provide valuable guidance for future research in the field of antimicrobial peptide design.

## Supporting information

**S1 Appendix. Details of model.**
(PDF)

**S2 Appendix. Basic analysis of AMPs.**
(PDF)

**S3 Appendix. Comparison of pre-trained and directly trained models.**
(PDF)

**S4 Appendix. Training details.**
(PDF)

**S5 Appendix. Physicochemical properties setting.**
(PDF)

**S6 Appendix. Docking score.**
(PDF)

## Author contributions

**Conceptualization:** Weizhong Zhao.

**Investigation:** Weizhong Zhao, Kaijieyi Hou, Chang Tang.

**Methodology:** Weizhong Zhao.

**Validation:** Kaijieyi Hou, Chang Tang, Jinlin Liu, Xiaohua Hu.

**Visualization:** Kaijieyi Hou, Yiting Shen.

**Writing – original draft:** Kaijieyi Hou.

**Writing – review & editing:** Chang Tang, Yiting Shen, Jinlin Liu, Xiaohua Hu.

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
