## [Decision Letter · Decision Letter 0]

29 Oct 2025

PCOMPBIOL-D-25-01993

A Novel Generative Framework for Designing Pathogen-targeted Antimicrobial Peptides with Programmable Physicochemical Properties

PLOS Computational Biology

Dear Dr. Zhao,

Thank you for submitting your manuscript to PLOS Computational Biology. After careful consideration, we feel that it has merit but does not fully meet PLOS Computational Biology's publication criteria as it currently stands. Therefore, we invite you to submit a revised version of the manuscript that addresses the points raised during the review process.

Please submit your revised manuscript within 60 days (December 25, 2025 at 23:59). If you will need more time than this to complete your revisions, please reply to this message or contact the journal office at ploscompbiol@plos.org. Please include the following items when submitting your revised manuscript:

We look forward to receiving your revised manuscript.

Kind regards,

Mohammad Sadegh Taghizadeh, Ph.D.

Academic Editor

PLOS Computational Biology

James Faeder

Section Editor

PLOS Computational Biology

**Journal Requirements:**

1) Please ensure that the CRediT author contributions listed for every co-author are completed accurately and in full.At this stage, the following Authors/Authors require contributions: Weizhong Zhao, Xiaohua Hu, Yiting Shen, Kaijieyi Hou, and Jinlin Liu. Please ensure that the full contributions of each author are acknowledged in the "Add/Edit/Remove Authors" section of our submission form.The list of CRediT author contributions may be found here: https://journals.plos.org/ploscompbiol/s/authorship#loc-author-contributions 2) We ask that a manuscript source file is provided at Revision. Please upload your manuscript file as a .doc, .docx, .rtf or .tex. If you are providing a .tex file, please upload it under the item type u2018LaTeX Source Fileu2019 and leave your .pdf version as the item type u2018Manuscriptu2019. 3) Please provide an Author Summary. This should appear in your manuscript between the Abstract (if applicable) and the Introduction, and should be 150-200 words long. The aim should be to make your findings accessible to a wide audience that includes both scientists and non-scientists. Sample summaries can be found on our website under Submission Guidelines:https://journals.plos.org/ploscompbiol/s/submission-guidelines#loc-parts-of-a-submission 4) Your manuscript is missing the following section heading: Abstract.  Please ensure all required sections are present and in the correct order. Make sure section heading levels are clearly indicated in the manuscript text, and limit sub-sections to 3 heading levels. An outline of the required sections can be consulted in our submission guidelines here:https://journals.plos.org/ploscompbiol/s/submission-guidelines#loc-parts-of-a-submission  5) Please upload all main figures as separate Figure files in .tif or .eps format. For more information about how to convert and format your figure files please see our guidelines: https://journals.plos.org/ploscompbiol/s/figures 6) Please revise your current Competing Interest statement to the standard "The authors have declared that no competing interests exist." 7) We note that the Data Availability Statement mentioned in the manuscript is different from that provided in the online submission form. The Data Availability statement  in the online submission form is currently as follows: 'All relevant data are within the manuscript and its Supporting Information files.' While the one in the manuscript states 'The data and source codes are available in GitHub at https://github.com/David-WZhao/AMPGen.' Please provide the complete Data Availability statement in the submission form and ensure that it matches the one mentioned in the manuscript..  Note: If the reviewer comments include a recommendation to cite specific previously published works, please review and evaluate these publications to determine whether they are relevant and should be cited. There is no requirement to cite these works unless the editor has indicated otherwise.

**Reviewers' comments:**

Reviewer's Responses to Questions

**Comments to the Authors:**

**Please note that two reviews are uploaded as attachments.**

Reviewer #1: This is an interesting and well-written study that introduces a new deep learning framework to design pathogen-targeted AMPs with programmable physicochemical properties. The combination of a CVAE with a diffusion model is technically sound, and the results are promising. The manuscript is clearly within scope and generally well organized, but several areas could be improved to make the work more accessible and convincing.

Major comments:

1. The authors should add a short paragraph that explicitly claim the novelty of this paper.

2. The authors rely on predicted MICs, toxicity scores, and docking simulations, which can all be biased by the datasets used to train those predictors. A short discussion paragraph on these in-silico limitations should be added in the discussion section.

3. In section 2.1.2, The criteria for “positive” vs. “negative” MIC thresholds (<10 µM and >20 µM) should be justified or referenced.

4. In Section 3.2 or Section 3.3, the authors should include a simple measure of novelty (maybe average percent identity of generated peptides to their nearest training sequence or n-gram overlap?). This will confirm the model produces new AMPs rather than variants of known ones.

5. In Figure 5D, why does increasing alpha-helicity and aromaticity boost antimicrobial activity? A brief biological rationale should be added explaining this trend.

6. In Section 3.3, the authors state isoelectric point behaves differently from other properties. The authors should include a short interpretation to show that they have considered this anomaly.

7. In Section 2.3.1, the text introduces “alpha” in Equation 2 as part of the noise schedule, but it is not linked to the later use of “alpha_t” and “alpha_(t–1)” in Algorithm 1. In addition, “beta_t” and “sigma_t” first appear in Algorithm 1 (Section 2.6) without any prior definition. The authors should explicitly define these schedules and parameters.

Minor comments:

1. Replace “AMPs treating E. coli” with “AMPs targeting E. coli” throughout

2. For Figures 4 and 5, add the number of generated peptides per condition and a short description of what each panel compares.

3. For Figures S8 and S9, briefly restate what docking and confidence scores mean.

Reviewer #2: The review is uploaded as an attachment

Reviewer #3: • Loss function lacks weighting coefficients:

In section 2.2.2. and Eq.1., the total loss combines KL divergence, reconstruction loss, and attribute preservation loss without any weighting factors. In practice, these terms differ in scale and importance; the absence of weighting or explanation raises concerns about training stability and reproducibility.

• Insufficient architectural details of key modules (BERT, fusion, condition embedding):

The manuscript does not provide adequate information about the BERT-based diffusion model (model size, pre-training, integration with condition vector) or the fusion layers in the CVAE. These omissions limit reproducibility.

• Fairness of MIC predictor comparison is questionable:

The authors’ MIC predictor is bacteria-specific, whereas the baseline predictors (Table 2) appear to be generic. This specialization likely provides an inherent advantage and should be either justified explicitly or controlled by training baseline models per bacterium.

• Limited assessment of novelty and diversity of generated peptides:

While biological activity is evaluated, the manuscript does not report sequence similarity metrics, novelty percentages, or diversity analyses. Without these, it is unclear whether the model generates substantially new peptides or variants of known sequences.

• Generalization beyond two bacterial species is untested:

The framework is claimed to be broadly applicable, but only E. coli and S. aureus are used. It remains unclear whether the model can generalize to other pathogens or requires re-training from scratch.

• Terminology consistency:

Terms such as “attribute,” “property,” and “feature” are used interchangeably. Consistent terminology would improve clarity.

• Details of t-SNE visualization missing:

Hyperparameters (e.g., perplexity, number of iterations) are not reported, which affects interpretability of the latent space plots.

• Segmentation procedure for UniProt sequences not fully described:

The method of segmenting proteins into short sequences (e.g., stride, overlap) should be specified.

Reviewer #4: Zhao et al. introduces a two-stage generative framework for designing antimicrobial peptides (AMPs) that are tailored to specific pathogens with desirable physicochemical properties. The approaches creatively combine CVAE for sequence generation and conditional diffusion model for fine-tuned on specific pathogens, plus MIC predictor to evaluate antimicrobial efficacy. Overall, the work is well-organized, technically solid. However, several aspects could be improved:

Major issues:

1. In 2.2.2. loss functions of CVAE, three items were combined into Eq.1. However, appears that no weights were applied to items. Did the authors use any weights for loss functions to balance the training? If yes, please report them; if not, please justify why equal weighting worked and how you tuned it.

2. In 3.2 model evaluation: the evaluation uses the authors’ MIC predictor to score all models’ outputs. This is reasonable but also potentially introduces bias if the predictor has learned patterns more aligned with the proposed model’s outputs. The authors should clarify how baseline sequences were obtained and evaluated. In addition, they compare MIC predictor performance to others in Table 2, showing slightly better results in MSE/MAE than ESKAPEE-MIC. Could you explain why the performance are so similar and discuss what strategies could be utilized to improve the performance.

3. I am wondering the novelty of star candidate AMPs designed by the framework. The authors may inspect their similarity with known AMPs.

Minor issues:

1. In 2.5. “contrastive fine-tuning is performed…” what is “contrastive fine-tuning”? No contrastive objective is defined.

2. how many sequences were generated per model for Fig. 4 and Table 1.

3. Model: specify the system size, CPU/GPU counts and runtime for both per step and total.

4. Figure 5: no y label. Add detailed description for each panel. In addition, please re-organize Fig 5 in the main text because Fig 5E occurs earlier than 5C.

5. Page 17. “Interestingly, regardless of how to set the targeted attribute of Isoelectric Point, this property distribution remains quite even, which may require further investigation.” Please discuss the reasons in Discussion.

6. Page 20. “This concentrated distribution allows the generative model to stably sample from this region, producing more high-antimicrobial activity AMPs with similar properties and functions. In contrast, the distribution of negative samples in the latent space exhibits greater dispersion. This suggests that the latent representation of negative samples is more complex, potentially related to diverse noise patterns or variations in features. Compared to the high-density clustering of positive samples,the more dispersed nature of negative samples presents a higher challenge for the generative model when treating AMPs with low antimicrobial activity. This di�erence further highlights the generative model’s focus on the generation direction of positive samples, optimizing its ability to generate AMPs with high-antimicrobial activity.” This paragraph should move to Discussion.

7. Fig 7. Legend. “In vitro test” should be changed to “In silico test” because only simulation experiments were performed.

8. Provide more methodlogical details about docking and MD simulations (e.g., systems construction, force field, running parameters and time).

Reviewer #5: This manuscript by Zhao and colleagues introduces an innovative generative framework for the design of antimicrobial peptides (AMPs) with tunable physicochemical properties and pathogen-specific targeting. The approach combines a conditional Variational Autoencoder (CVAE) with a conditional diffusion model, further supported by a MIC predictor. The framework is rigorously evaluated through simulation experiments, visual analyses, and in silico screening, demonstrating notable improvements over existing methods in generating potent, low-toxicity AMPs targeting E. coli and S. aureus.

Overall, the study presents a compelling and technically robust methodology with significant potential to advance antimicrobial drug discovery. With further experimental validation and expansion to a broader range of pathogens, this framework could become a foundational tool for next-generation AMP development. I have the following comments:

Major comments:

1. The results presented are comprehensive and well-supported by in silico analyses, however, the manuscript does not mention any experimental validation (e.g., antimicrobial assays, hemolysis tests). Including wet-lab validation for at least one generated AMP would significantly enhance the translational relevance and practical impact of the study.

2. The model appears to be trained primarily on E. coli and S. aureus, with limited representation of other bacterial species. Expanding the training dataset to include a broader range of pathogens, particularly non-ESKAPE organisms, would improve generalizability. Alternatively, a discussion of potential transfer learning strategies could be beneficial.

3. The manuscript does not address the framework’s applicability to longer peptides or non-peptide antimicrobial agents. A brief discussion on the potential limitations or extensions regarding peptide length and structural diversity would help clarify the scope of the approach.

4. The model seems to struggle with controlling the isoelectric point of generated sequences. It may be helpful to investigate whether this limitation arises from the underlying data distribution, encoding strategy, or model architecture. Incorporating a regularization term or exploring alternative encoding methods could improve control over this property.

5. It is not entirely clear which core datasets from UniProt were used in this study. UniProt provides several distinct resources, including UniProtKB for protein sequences, Proteomes for species-level data, UniRef for protein clustering, and UniParc as a sequence archive. Additionally, UniProt distinguishes between reviewed (Swiss-Prot) and unreviewed (TrEMBL) entries, which vary in terms of curation and redundancy. Clarifying which subset was utilized, particularly whether reviewed or unreviewed data were included, would be helpful. If unreviewed data were used, please describe how redundancy and annotation quality were addressed. It would also be valuable to comment on whether the UniProt and GRAMPA datasets employed in this study adequately represent the diversity of real-world antimicrobial peptides across different species and environments.

6. The proposed framework demonstrates impressive performance, but AI-based generative models like the CVAE and diffusion components used here are inherently black-box systems. I encourage the authors to provide additional interpretability insights, such as feature attribution, latent space diagnostics, or decision rationale from the MIC predictor. This would enhance transparency and trust, especially for clinical or experimental applications where explainability is critical.

7. While docking scores below –300 and confidence scores above 0.96 are promising, it would be helpful to include comparative benchmarks or reference peptides to contextualize these values.

8. Consider summarizing key findings from Tables S2–S4 in a graphical format (e.g., radar plots or heatmaps) to enhance interpretability of attribute differences across AMP variants.

Minor comments:

1. In the Introduction

a. Provide the full form of GRAMPA?

b. The authors mentioned “we first sample the noise data”- Could they clarify what kind of noise is meant in this context?

2. Ensure consistent formatting of units.

3. Clarify the role of the MIC predictor in generation vs. evaluation.

4. Include more details on the docking scoring functions and thresholds used.

5. To support reproducibility and facilitate further research, please consider including the link to your implementation in the abstract. For example: "The source code for our framework is available at https://github.com/David-WZhao/AMPGen." This would make it easier for readers to access and explore your work directly.

**Have the authors made all data and (if applicable) computational code underlying the findings in their manuscript fully available?**

Reviewer #1: Yes

Reviewer #2: Yes

Reviewer #3: **No: **The manuscript does not provide any links to code repositories (e.g., GitHub) or data sources beyond general references to UniProt and GRAMPA.

Reviewer #4: Yes

Reviewer #5: Yes

PLOS authors have the option to publish the peer review history of their article (what does this mean?). If published, this will include your full peer review and any attached files.

Reviewer #1: No

Reviewer #2: No

Reviewer #3: No

Reviewer #4: No

Reviewer #5: **Yes: **Dr. Reema Singh

**Figure resubmission:**
---

## [Decision Letter · Decision Letter 1]

28 Nov 2025

PCOMPBIOL-D-25-01993R1

A Novel Generative Framework for Designing Pathogen-targeted Antimicrobial Peptides with Programmable Physicochemical Properties

PLOS Computational Biology

Dear Dr. Zhao,

Thank you for submitting your manuscript to PLOS Computational Biology. After careful consideration, we feel that it has merit but does not fully meet PLOS Computational Biology's publication criteria as it currently stands. Therefore, we invite you to submit a revised version of the manuscript that addresses the points raised during the review process.

We look forward to receiving your revised manuscript.

Kind regards,

Mohammad Sadegh Taghizadeh, Ph.D.

Academic Editor

PLOS Computational Biology

James Faeder

Section Editor

PLOS Computational Biology

**Reviewers' comments:**

Reviewer's Responses to Questions

**Comments to the Authors:**

Reviewer #1: I thank the authors for revising the manuscript and addressing my comments. The revised manuscript is now suitable for publication.

Reviewer #2: All of my comments from the first review have been addressed by the authors. Especially section A in the SI makes it much more accessible to a broader audience, that could be interested in it.

There are only two minor suggestions, that I still have.

1. The authors explained to me in their response what ESM and MLP (Fig 3 and subsection 2.4.1) means. I think it would improve this section, if you just include the full name of these abbreviations als in the figure caption or in the text

2. It is clear from the text what is shown in tables and figures. However, adding some of the information also in the captions (f.e. Table 4, that the numbers that are shown there are proportions) could improve the accessibility even further.

Overall, I think the quality of this article improved significantly, and it could provide an interesting and promising new approach.

Reviewer #3: First, I would like to thank you for the revisions made so far, especially for adding the novelty analysis and for expanding the explanation of the segmentation procedure. However, there are still several important points that require clarification and further refinement in order to ensure full transparency.

Comment 1:

Regarding the novelty and diversity analysis, although the inclusion of Table 2 and the 6-gram and 4-gram novelty scores is appreciated, I strongly recommend presenting this part as a separate subsection so that the metrics and methodology can be described in a complete and self-contained manner. At the moment, the definition of the 6-gram and 4-gram overlap remains unclear. It is not specified what the overlap is being measured against: the reference AMP database?, the training set?, sequences generated by other models?, or some other source??. Without explicitly defining the reference set, the novelty values cannot be properly interpreted. Additionally, the manuscript does not indicate how many 6-grams and 4-grams were examined, how the overlap was normalized, or whether novelty is defined as 1 − overlap or based on another formulation. Because peptide sequences are relatively short, extremely high novelty values such as 0.9998 for 6-grams require a clear methodological explanation to confirm that the computation is sound.

Furthermore, diversity has not been evaluated, even though diversity is just as important as novelty in generative peptide research. Novelty measures how different generated peptides are from known sequences, whereas diversity measures how different the generated peptides are from one another. Without assessing diversity, it is impossible to determine whether the model is producing a wide range of structurally and physicochemically distinct peptides or merely generating small variations of a limited set of motifs. Diversity can be quantified using standard metrics such as sequence entropy, pairwise distance distributions, clustering-based diversity, or physicochemical diversity across generated samples. Including such analyses is essential, because high novelty alone can even result from generating random or biologically implausible sequences, which may not reflect true generative capability. For the comparisons in Table 2, additional methodological clarity is needed. It should be stated how many sequences each model generated, whether the sequence lengths were controlled and matched across methods, and whether the same sampling procedures were used. Without having these conditions specified, the comparisons may be difficult to interpret reliably.

Comment 2:

With respect to the segmentation procedure, I appreciate that additional explanation has been added, but several key details are still missing. In particular, the stride used for the sliding window has not been specified. It remains unclear whether the stride was fixed, equal to one, proportional to the window size, or randomly chosen. The stride directly affects the density of extracted fragments and the distribution of n-grams, so it must be clearly defined for the segmentation to be reproducible.

Similarly, the balancing strategy for ensuring an equal distribution of segment lengths has not been described with sufficient precision. The manuscript mentions stopping once a “target count” for each length is reached, but the basis for selecting these target counts is not explained. It is important to clarify whether these targets were derived from actual AMP length distributions, set uniformly across lengths, or chosen using another criterion. Additionally, since the window size is randomly selected between 2 and 50, there is a risk of introducing bias toward certain lengths unless this process is carefully controlled. The manuscript does not currently explain what mechanisms, if any, were used to prevent such bias or to ensure that shorter or longer fragments were not disproportionately represented.

Reviewer #4: All my concerns have been well addressed. I recommend it to be published in the journal.

Reviewer #5: Thank you for the revised manuscript. I appreciate that the authors have carefully addressed my comments

**Have the authors made all data and (if applicable) computational code underlying the findings in their manuscript fully available?**

Reviewer #1: Yes

Reviewer #2: Yes

Reviewer #3: Yes

Reviewer #4: Yes

Reviewer #5: Yes

PLOS authors have the option to publish the peer review history of their article (what does this mean?). If published, this will include your full peer review and any attached files.

Reviewer #1: No

Reviewer #2: No

Reviewer #3: No

Reviewer #4: No

Reviewer #5: **Yes: **Dr. Reema Singh

**Figure resubmission:**
---

## [Decision Letter · Decision Letter 2]

10 Dec 2025

Dear Dr. Zhao,

We are pleased to inform you that your manuscript 'A Novel Generative Framework for Designing Pathogen-targeted Antimicrobial Peptides with Programmable Physicochemical Properties' has been provisionally accepted for publication in PLOS Computational Biology.

Best regards,

Mohammad Sadegh Taghizadeh, Ph.D.

Academic Editor

PLOS Computational Biology

James Faeder

Section Editor

PLOS Computational Biology

Reviewer's Responses to Questions

**Comments to the Authors:**

Reviewer #2: I thank the authors for revising the manuscript and address all my comments. From my perspective, the manuscript is now ready for publication.

Reviewer #3: All my concerns have been well addressed. I recommend it to be published in the journal.

**Have the authors made all data and (if applicable) computational code underlying the findings in their manuscript fully available?**

Reviewer #2: Yes

Reviewer #3: Yes

PLOS authors have the option to publish the peer review history of their article (what does this mean?). If published, this will include your full peer review and any attached files.

Reviewer #2: No

Reviewer #3: No

---

## [Editor Report · Acceptance letter]

PCOMPBIOL-D-25-01993R2

A Novel Generative Framework for Designing Pathogen-targeted Antimicrobial Peptides with Programmable Physicochemical Properties

Dear Dr Zhao,

I am pleased to inform you that your manuscript has been formally accepted for publication in PLOS Computational Biology. Your manuscript is now with our production department and you will be notified of the publication date in due course.

With kind regards,

Narmatha Raju, M.Sc
